# Preferential molecular recognition of heterochiral guests within a cyclophane receptor

Manuel Weh[1], Kazutaka Shoyama [2] & Frank Würthner [1,2] ✉

The discrimination of enantiomers by natural receptors is a well-established phenomenon. In contrast the number of synthetic receptors with the capability for enantioselective molecular recognition of chiral substrates is scarce and for chiral cyclophanes indicative for a preferential binding of homochiral guests. Here we introduce a cyclophane composed of two homochiral core-twisted perylene bisimide (PBI) units connected by *p*-xylylene spacers and demonstrate its preference for the complexation of [5]helicene of opposite helicity compared to the PBI units of the host. The pronounced enantio-differentiation of this molecular receptor for heterochiral guests can be utilized for the enrichment of the *P*-PBI-*M*-helicene-*P*-PBI epimeric bimolecular complex. Our experimental results are supported by DFT calculations, which reveal that the sterically demanding bay substituents attached to the PBI chromophores disturb the helical shape match of the perylene core and homochiral substrates and thereby enforce the formation of syndiotactic host-guest complex structures. Hence, the most efficient substrate binding is observed for those aromatic guests, e. g. perylene, [4]helicene, phenanthrene and biphenyl, that can easily adapt in non-planar axially chiral conformations due to their inherent conformational flexibility. In all cases the induced chirality for the guest is opposed to those of the embedding PBI units, leading to heterochiral host-guest structures.

Narcissistic self-sorting, is a well-known phenomenon for supramolecular entities[1,2] and can in most cases be explained by a better shape complementarity between the corresponding molecules of the same configuration, leading to less steric hindrance[3] or increased intermolecular interactions, e.g., dispersion interactions[4-10] or other driving forces[11-13]. In contrast, if enantiomers prefer to bind their mirror image, this is called chiral self-discrimination or heterochiral self-sorting[14-19]. In the case of helicene structures, homochiral self-recognition is accepted to be a general trend in noncovalent bonding events[20,21]. In this context, the homochiral supramolecular complexes observed upon encapsulation of [4]- and [5]-carbohelicenes within an inherently chiral PBI cyclophane introduced recently by us extends this empirical result to non-identical molecules with structural analogies by revealing the perfect accommodation of substrates with the same configuration as the adjacent PBIs within a tailored cavity[22].

Such perfect shape matching between a substrate and a cavity supports the common view of receptor-substrate fit in nature that is still often dominated by the picture of tailored cavities for substrate complexation with a perfect shape complementarity between host and guest, leading to efficient binding. This rationale is also present in the three-point attachment (TPA) model, which is applied to explain stereospecific binding in natural systems by a shape match of substrate and the active binding site of a receptor[23]. However, taking conformational flexibility of the system into account is of paramount

[1]Institut für Organische Chemie, Universität Würzburg, Am Hubland, 97074 Würzburg, Germany. [2]Center for Nanosystems Chemistry & Bavarian Polymer Institute, Universität Würzburg, Theodor-Boveri-Weg, 97074 Würzburg, Germany. ✉e-mail: wuerthner@uni-wuerzburg.de

importance in natural systems as conformational changes in protein-substrate assemblies are responsible for allosteric effects that contribute quite significantly to the regulation of biochemical processes[24,25]. Besides, it is also well known that many artificial complexes, held together by noncovalent interactions, can not be described with rigid geometries of receptor and guest structures. Accordingly, the relevance of structural adaption upon substrate binding is highly topical in the field of supramolecular host-guest chemistry[26–30]. For artificial receptors whose guest complexation affinity is mainly driven by π−π interactions[31], i.e., dispersion and electrostatic interaction, an empirical dependence of the binding strength on the size of the interacting π-surfaces, often related to the number of π-electrons provided by the substrate, has been observed for several examples[32–37]. Nevertheless, the number of π-electrons is not always a meaningful parameter with regard to complexation strength if the π-scaffolds are distorted.

Here, we present a chiral PBI host endowed with highly distorted aromatic π-surfaces that hamper the host from the efficient noncovalent binding of flat PAH guest molecules. Therefore, for this host, the number of π-electrons that a substrate offers is not a meaningful parameter to predict the binding strength and the guest flexibility and conformational adaptability need to be taken into account. As a more important consequence, the structural influence of the bay substituents prevails over the shape match of the helically twisted perylene and the guest, enabling a pronounced preference for the molecular recognition of heterochiral aromatic guest molecules.

## Results

### Synthesis and characterization

The synthesis of the cyclophane rac-1 (Fig. 1) begins with the distortion of a PBI chromophore into a chiral π-scaffold. Besides the diagonal or lateral bridging of the 1,7[4,5,22,38] or the 1,12[39,40] bay area positions, respectively, another way to achieve helically twisted PBIs[41] is the steric

overcrowding by bulky substituents in the bay position[42–45]. Accordingly, we started with the fourfold arylation of a PBI in the bay position and a subsequent saponification, which yields the literature known perylene bisanhydride rac-2 as a mixture of atropo-enantiomers[46]. Imidization of bisanhydride rac-2 with mono Boc-protected p-xylylenediamine resulted in the corresponding perylene bisimide derivative rac-3b. Thereafter, deprotection of the Boc-protecting group of compound rac-3b and macrocyclization with rac-2 gave access to rac-1 as a mixture of 1-PP and 1-MM. Notably, no 1-PM isomer could be detected after the macrocyclization step, which might be explained by a repulsive interaction between the bay substituents in the transition state and therefore a preferred homochiral stacking within the cyclophane host. 2D NMR studies were performed in order to assign the proton signals (Supplementary Figs. 11–13 and text below). The proton signals give, in accordance with the structural properties of the macrocycle, two sets for the perylene protons (Fig. 1d) as well as for the substituents: one set for the protons that point towards the cavity and one for the more distant protons. In contrast, the monomeric reference shows only one set of signals.

Chiral resolution by HPLC (Supplementary Fig. 1) worked efficiently for this system with a baseline separation already in the first cycle, thereby affording the two enantiopure homochiral cyclophanes with an enantiomeric excess of >99% as proven by analytical chiral HPLC. The absolute configuration of the cyclophane could subsequently be assigned by TD-DFT calculations (Supplementary Fig. 17). In addition, we have also synthesized the monomeric reference compound rac-3a. For details on the experimental procedures and characterization of all new compounds see the Supplementary Information.

### Structural and (chiro-)optical properties

To get further structural insights beyond those from NMR experiments we grew a single crystal suitable for X-ray analysis of the cyclophane

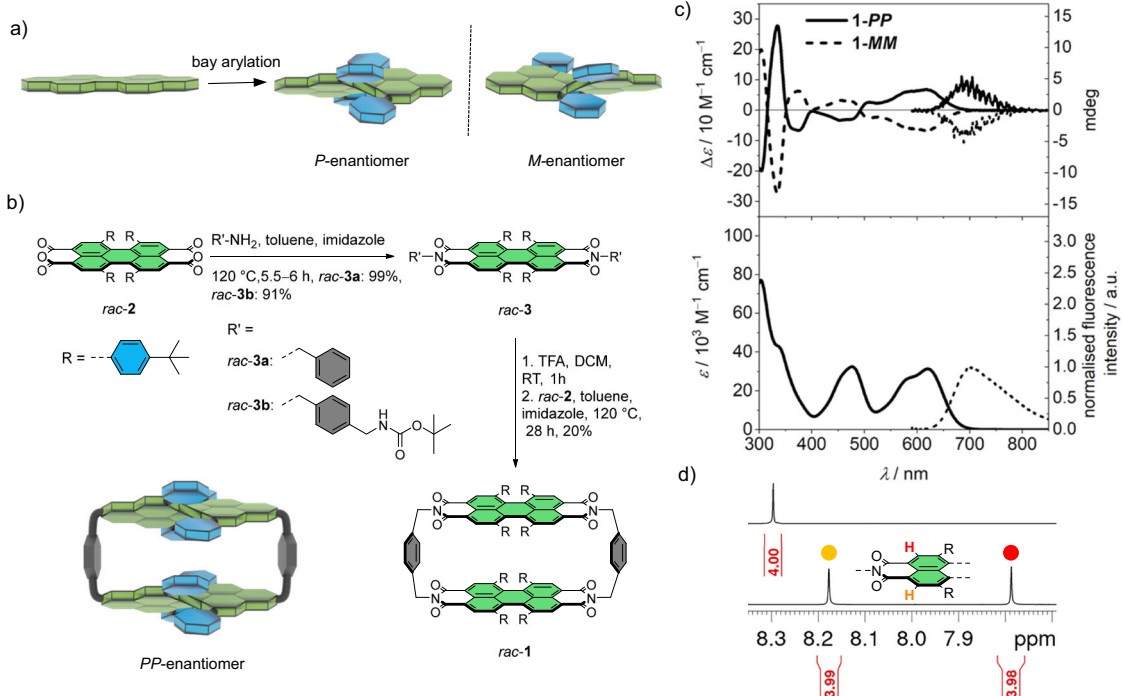

**Fig. 1 | Synthesis and properties of chiral PBI cyclophane. a** General concept of introducing helical chirality to PBIs by bay-arylation. **b** Synthesis of racemic perylene bisimide cyclophane rac-1 and monomeric reference rac-3a and schematic depiction of the PP-cyclophane. **c** Circular dichroism and CPL spectra of 1-PP and 1-MM enantiomers of the cyclophane (top) and UV/vis absorption and fluorescence spectra of the racemic mixture of cyclophane (bottom) in chloroform. **d** Excerpt from 400 MHz ¹H NMR spectra of rac-3a and rac-1 in CDCl₃ at 295 K.

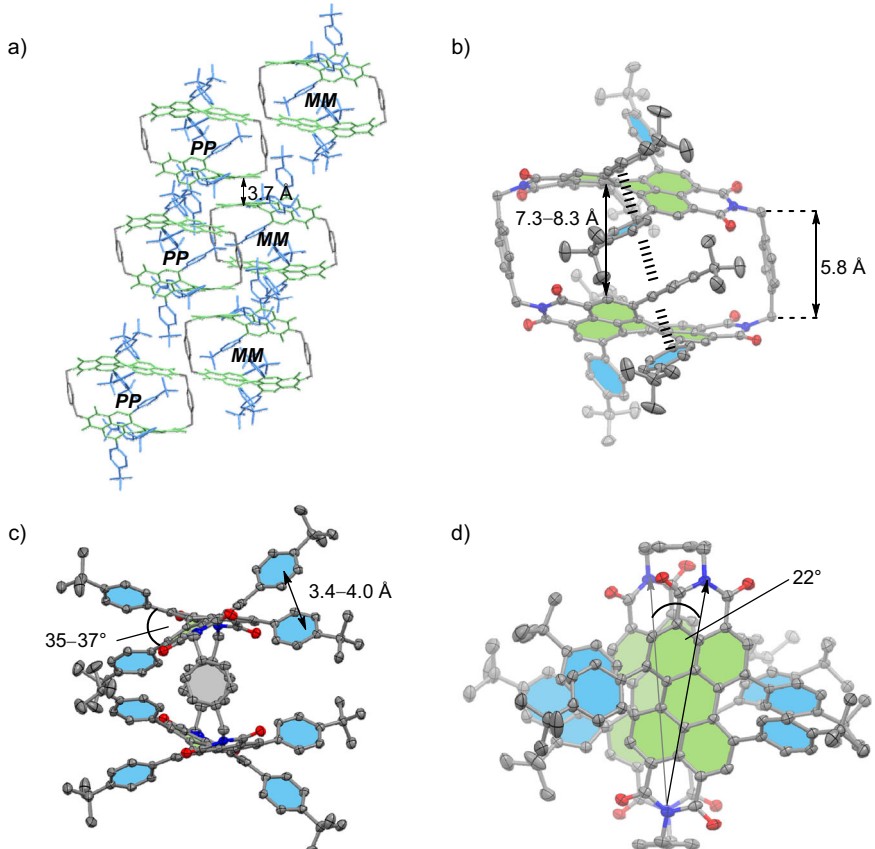

**Fig. 2 | Single crystal X-ray analysis of *rac*-1.** Molecular structure and packing of cyclophane *rac*-1 in the solid state according to single crystal X-ray analysis. **a** Packing arrangement in the crystal is shown in the side view. **b** Side, (**c**) front and (**d**) top view of the molecular structure of **1-PP**. Hydrogen atoms and solvent molecules are omitted for clarity. The perylene units are highlighted in green, the bay substituents in blue and the spacer unit in grey (the thermal ellipsoids are set to 50% probability).

*rac*-1 by slow diffusion of methanol into the chloroform solution (Fig. 2). The molecule crystallizes in the C2/c space group with four cyclophanes per unit cell. The cavity is filled with chloroform molecules and the PBI moieties of adjacent cyclophane molecules pack in a heterochiral fashion with a distance of 3.7 Å between the chromophores of one **1-PP** and one **1-MM** isomer without π-contact (Fig. 2a). The distance between the PBI units within one cyclophane is 7.3–8.3 Å. Accordingly, the choice of the *para*-xylylene spacer provides the expected perfect distance for the encapsulation of polycyclic aromatic hydrocarbons (Fig. 2b). From the X-ray structure, the high steric demand of the *tert*-butyl-phenyl substituents becomes apparent as well as the concomitant rigidity of the whole molecule. The close proximity of the phenyl groups leads not only to high twist angles between the naphthalene subunits of 35–37° (Fig. 2c) but also supports π−π interactions between the adjacent phenyl moieties, leading to a quadruple π−stack (Fig. 2b). As a result of the steric overcrowding and influenced by the size and shape of the respective guests (here solvent molecules, for specific guests see *vide infra*), the whole cyclophane has a distorted symmetry-broken geometry (point group: $C_2$), leading to a rotational offset of the long axis between the chromophores of approximately 22° in the solid state (Fig. 2d).

The optical properties of the cyclophane were studied by UV/vis and circular dichroism (CD) absorption as well as fluorescence and circularly polarized luminescence (CPL) spectroscopies (Fig. 1c). For specific details on the optical properties we refer to the Supplementary Information and summarize here only the major results. Thus, we observe several absorption bands in the UV/vis spectral range and only weak solvatochromism (Supplementary Fig. 14c). The vibronic fine structure of the $S_0 \rightarrow S_1$ absorption band at ~600 nm is less pronounced

than for other bay substituted perylene bisimide cyclophanes[22,35,47] and the absorbance of the $S_0 \rightarrow S_2$ band is rather strong which could be rationalized by a significant involvement of the phenyl substituents in this transition as supported by TD-DFT calculations (Supplementary Fig. 18). The fluorescence properties are more dependent on the solvent polarity with lower quantum yield ($\phi_f = 0.23$) and lifetime ($\tau = 11.5$ ns) in chloroform that increase in less polar solvents such as tetrachloromethane and methylcyclohexane (Supplementary Fig. 15, Supplementary Table 1). In the visible range, the CD absorption complies well with the UV/vis absorption with two broad monosignated signals for the $S_0 \rightarrow S_1$ and the $S_0 \rightarrow S_2$ transitions with opposite sign (400−700 nm). Importantly, the naphthalene-related absorption in the UV regime is in agreement with our assignment of the enantiomers by exciton chirality method[48]. Accordingly, a positive CD exciton couplet with a zero crossing at $\lambda = 316$ nm is apparent in the CD spectrum of **1-PP**, indicating a clockwise stack of the naphthalene subunits when looking from the short side of the cyclophane. The corresponding enantiomer has the expected mirror image CD spectrum. The absorption dissymmetry factor is in the medium range for small organic molecules with $g_{abs} = \Delta\varepsilon/\varepsilon = 2.1 \times 10^{-3}$ (at $\lambda = 620$ nm). The CPL spectra exhibit the expected mirror image of the CD spectrum for the lowest energy transition with $g_{lum}$ values of $2.1 \times 10^{-3}$ and $-1.7 \times 10^{-3}$ for the two enantiomers, respectively (at $\lambda = 675$ nm, Supplementary Fig. 14b).

### Complexation of non-chiral guest molecules

The suitability of this cyclophane host as a molecular receptor for the complexation of guest molecules was studied by UV/vis and NMR titration experiments. As the guest complexation within PBI

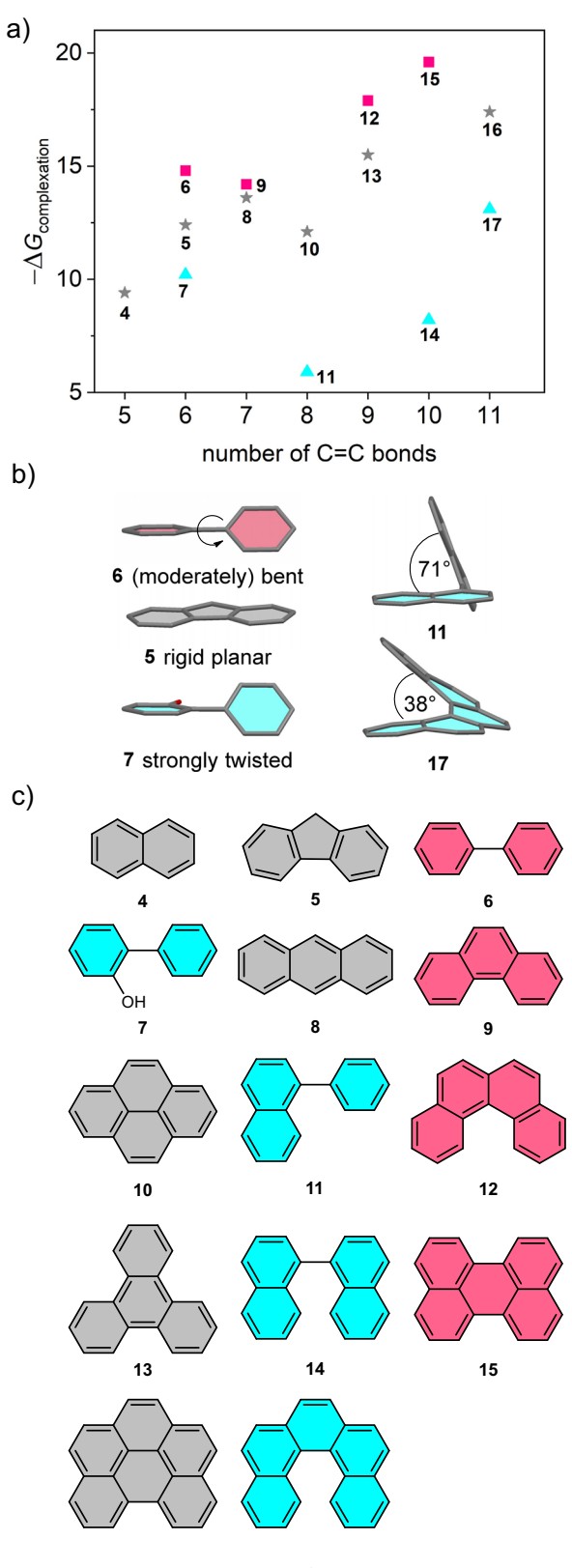

**Fig. 3 | Binding studies. a** Plot of Gibbs free complexation energies at 295 K versus the number of C = C double bonds of the guests' π-scaffold. The guests are classified in three groups: guests with a moderately bent or easily bendable structure that can adapt to the receptor (red), guests with a rigid planar structure (grey) and guests with a strongly twisted three-dimensional structure (blue). **b** Energy minimized structures by DFT of selected guest molecules and (**c**) chemical structure of the substrates which were employed as guests in the titration studies.

**Table 1 | Binding constants for the complexation of PAHs with *rac*-1 determined by UV/vis or ¹H NMR titration experiments in tetrachloromethane at 295 K and Gibbs free energies for the corresponding complex formation**

| guest | $(K_a \pm \Delta K_a)$[a)] [$M^{-1}$] | $-\Delta G^0$ [kJ $mol^{-1}$][b)] |
|---|---|---|
| perylene | $(2.9 \pm 0.1) \times 10^3$ | 19.6 |
| [4]helicene | $(1.47 \pm 0.04) \times 10^3$ | 17.9 |
| benzo[*ghi*]perylene | $(1.15 \pm 0.03) \times 10^3$ | 17.3 |
| triphenylene | $(5.5 \pm 0.5) \times 10^2$ | 15.5 |
| 1,1'-biphenyl | $(4.1 \pm 0.1) \times 10^2$ | 14.8 |
| phenanthrene | $(3.3 \pm 0.4) \times 10^2$ | 14.2 |
| anthracene | $(2.6 \pm 0.1) \times 10^2$ | 13.6 |
| *rac*-[5]helicene[c)] | $(2.1 \pm 0.1) \times 10^2$ | 13.1 |
| fluorene | $(1.6 \pm 0.1) \times 10^2$ | 12.4 |
| pyrene | $(1.4 \pm 0.1) \times 10^2$ | 12.1 |
| 2-hydroxybiphenyl | $64 \pm 7$ | 10.2 |
| naphthalene | $47 \pm 5$ | 9.4 |
| 1,1'-binapthyl[d)] | $28 \pm 4$ | 8.2 |
| phenylnapthalene[d)] | $11 \pm 1$ | 5.9 |

[a]The given error for $K_a$ is the analytical error from the local 1:1 binding fit. [b]Calculated with $\Delta G^0 = -RT \ln(K_a)$. The analytical error of $\Delta G^0$ is less than 1.5 kJ $mol^{-1}$ for our titration studies. [c]The titration studies of the corresponding enantiopure mixtures can be found in the Supplementary Information Fig. 42: $K_a = 220\ M^{-1}$ (heterochiral host-guest mixture) and $K_a = 46\ M^{-1}$ (homochiral host-guest mixture). [d]Determined by ¹H NMR titration in CCl₄/MCH-$d_{14}$ 3:1 (v:v).

cyclophanes is known to be mainly driven by π−π interactions[35,47], we selected initially a series of PAHs as guests that should fit into the cavity with a stepwise increase of π−electron count, starting with naphthalene (5 double bonds) until [5]helicene and benzo[*ghi*]perylene (11 double bonds, Fig. 3). The association constants were determined by nonlinear curve fitting of data from titration studies (Supplementary Figs. 25–38) and the results are summarized in Table 1. For the titration studies we chose tetrachloromethane as a solvent which yielded binding constants for the guest complexations up to $K_a = 2.9 \times 10^3\ M^{-1}$ for perylene encapsulation. On first glance, this strong complexation of perylene might be surprising. However, it should be noted that the central ring of perylene is less aromatic with long carbon-carbon bonds and that already the hydrogen substituents in the bay area of perylene suffer from some crowding. Thus, little energy is needed for a rotational twist of the two naphthalene subunits in a propeller-like structure for the perylene scaffold[49] (Supplementary Fig. 21).

Most of our complexation studies show a decrease in the absorption of the lowest energy PBI-related transition with a concomitant red shifted shoulder which indicates a charge transfer character of the complex. However, a plot of the corresponding Gibbs free energies (Fig. 3a) revealed that there is no simple correlation between the π-count of the guest, i.e. the number of carbon-carbon double bonds, and the binding affinity like for other cyclophanes[33–35]. Accordingly, the highly twisted and rigid conformation of this cyclophane leads to a certain guest specificity. Based on the structural properties of the guests, we can classify them into three groups. For guests with a planar rigid structure (marked in grey) we see a trend which shows the expected increasing binding affinity with a larger aromatic π-plane due to increased dispersion interactions between the guest and the PBI units. Only pyrene with its incongruous and stiff geometry compared to the perylene units of the neighbouring chromophores seems to somewhat counteract this trend, which has also been observed previously for cyclophanes composed of core-distorted PBIs[35,47]. In order to rule out external binding, we carried out a titration with reference compound *rac*-**3a**, revealing negligible changes in the UV/vis spectrum upon the addition of the largest guest

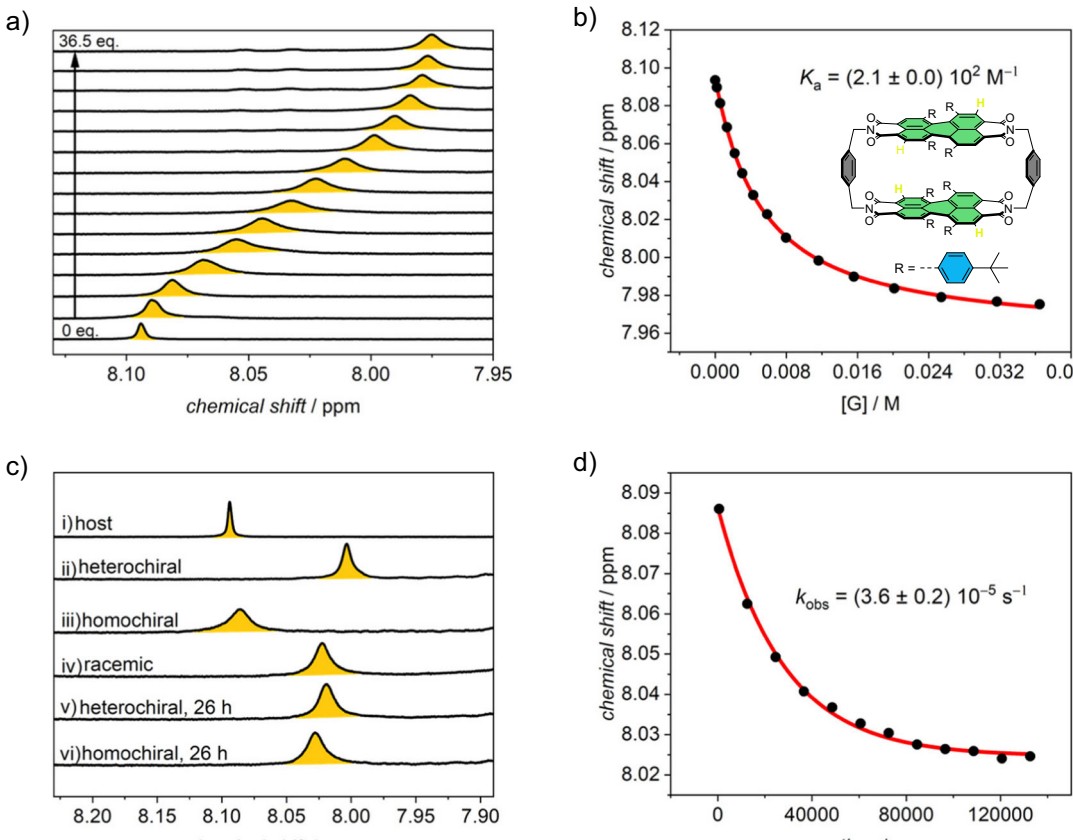

**Fig. 4 | Complexation studies with [5]helicene. a** Excerpt of $^1$H NMR (CCl$_4$/MCH-$d_{14}$ 3:1 (v:v), $c$ ($rac$-**1**) = 1.0 × 10$^{-3}$ M, 295 K) titration experiment of host $rac$-**1** with $rac$-[5]helicene and (**b**) the nonlinear curve fit of the obtained chemical shift of PBI ortho proton versus guest concentration according to the 1:1 binding model. **c** Excerpt of $^1$H NMR spectra (CCl$_4$/MCH-$d_{14}$ 3:1 (v:v), $c$ (host) = 1.0 × 10$^{-3}$ M, 295 K) of

different configurational combinations of the host and [5]helicene (5.8 eq.): (i) host, (ii) host+guest heterochiral, (iii) host+guest homochiral, (iv) host+guest racemic, (v) host+guest heterochiral after 26 h, (vi) host+guest homochiral after 26 h. **d** Plot of time-dependent chemical shift of the most downfield shifted PBI ortho proton of **1-PP** upon the addition of $P$-[5]helicene (5.8 eq.).

(Supplementary Fig. 44). For strongly twisted more three-dimensional guest molecules which have a restricted flexibility (marked in blue), like for example 2-hydroxybiphenyl, we determined significantly lower binding energies. Notably are in this context phenylnaphthalene and 1,1'-binaphthyl, whose binding by $rac$-**1** could only be detected at higher concentration in NMR experiments. Thus, the twist between the aromatic subunits in these molecules (71° for 1,1'-binaphthyl, see Fig. 3b) obviously does not match well within the available space of the cyclophane receptor's cavity. In contrast, [5]helicene, with a significantly lower angle of 38° is again well accommodated within the host. The third group of guest molecules (marked in red) shows the best binding properties and is characterized by planar or only moderately bent structures and some degree of conformational flexibility as before discussed for perylene. Thus, whilst perylene is assumed to be bent upon complexation, twisted molecules like 1,1'-biphenyl or [4] helicene can adapt to the cyclophane cavity by partial planarization, thereby enabling a more efficient binding. Hence, the strongly limited adaptability of PBI dyes with interlocked phenyl substituents and the resulting highly distorted nature of the host's cavity is the reason for a preferential binding of guest molecules that can optimize their shape match with the receptor.

## Complexation of Chiral guest molecules

The previous complexation studies suggested that cyclophane **1** has the highest binding affinity for non-planar guest molecules. As the helical twist of bay-substituted PBIs with their "phenanthrene unit" in the bay area is structurally related to carbohelicenes, the best shape

match might be expected for supramolecular inclusion complexes with homochiral helicene guest molecules. Although [5]helicene has a lower affinity for complexation by the cyclophane compared to [4] helicene, this substrate is particularly suitable for the following studies because it is characterized by a sufficiently high inversion barrier of about 101 kJ mol$^{-1}$[50]. This allows us to monitor the enantioselectivity of the molecular recognition by the host as well as the conversion of its isomeric forms on a reasonable time scale. Thus, we first carried out $^1$H NMR studies with either racemic or homo- and heterochiral mixtures of host and guest in a mixture of tetrachloromethane and methylcyclohexane-$d_{14}$ (3:1, v:v). Notably, UV/vis titration study in the same solvent mixture revealed almost no impact of the methylcyclohexane on the complexation thermodynamics compared to pure tetrachloromethane (Supplementary Fig. 39). The corresponding NMR titration (Fig. 4a, b and Supplementary Fig. 40) for our initial experiment with host $rac$-**1** with $rac$-[5]helicene in the solvent mixture shows a broadening of most host signals upon guest addition. However, the most strongly downfield shifted ortho perylene proton that points away from the cavity can be followed very well during the titration experiment and shows not only a distinct upfield shift upon guest addition but stays also sharp over the whole experiment. Subsequent nonlinear curve fitting of this proton signal reveals a binding constant of $K_a$ = 2.1 × 10$^2$ M$^{-1}$, confirming our data from UV/vis host guest titration studies (Table 1).

As we were particularly interested in the impact of the guest configuration on the binding affinity in epimeric host-guest complexes, we compared next the chemical shift of the selected ortho

perylene proton upon the addition of a particular amount of [5]helicene (5.8 equivalents) in different configurational combinations of host and guest (Fig. 4c). For the samples measured directly after dissolution of **1-PP** and the respective guests we see an upfield shift of 0.07 ppm if the racemic guest is added to the racemic host (Fig. 4c, iv). Unexpectedly, this upfield shift is with 0.09 ppm more pronounced if we add *M*-[5]helicene (Fig. 4c, ii) and less pronounced if we add *P*-[5]helicene (Fig. 4c, iii), indicating that for the cyclophane a preferential binding takes place for the guest with the opposite chirality compared to the configuration of the PBI units of the receptor. In the case of the homochiral mixture, the shift of the proton is indeed almost unchanged (upfield shift of only 0.01 ppm) but only the signal itself is broadened. However, after approximately one day, the signals of both combinations approach the chemical shift for the racemic case (Fig. 4c, iv–vi). A plot of the chemical shift over the time for the homochiral mixture and subsequent data fitting with the first-order-kinetics (Fig. 4d) reveals a rate constant of $k_{\mathrm{obs}} = 3.6 \times 10^{-5}\,\mathrm{s}^{-1}$, corresponding to a barrier of $\Delta G^{\ddagger} = 99.0\,\mathrm{kJ\,mol}^{-1}$ according to equation S2 and S3 (see Supplementary Information). This barrier is in accordance with the inversion barrier of [5]helicene[50,51] and was further confirmed by time-dependent CD measurement in the same solvent mixture in the absence of the host (Supplementary Fig. 42). Accordingly, we see an increasing amount, i.e., enrichment of the heterochiral complex upon enantiomerization of *P*-[5]helicene into *M*-[5]helicene in the presence of **1-PP**. UV/vis titration experiments confirm this observation, revealing significantly reduced optical changes upon the addition of the same amount of homochiral [5]helicene to the host compared to the heterochiral case (Supplementary Fig. 43). The corresponding binding constants amount to $K_{\mathrm{a}} = 220\,\mathrm{M}^{-1}$ for the heterochiral and $K_{\mathrm{a}} = 46\,\mathrm{M}^{-1}$ for the homochiral complex. From the difference in the binding constants of the two complexes, we deduced a difference in Gibbs free energy of 3.8 kJ mol⁻¹ for the complex formations of the two epimeric host-guest structures.

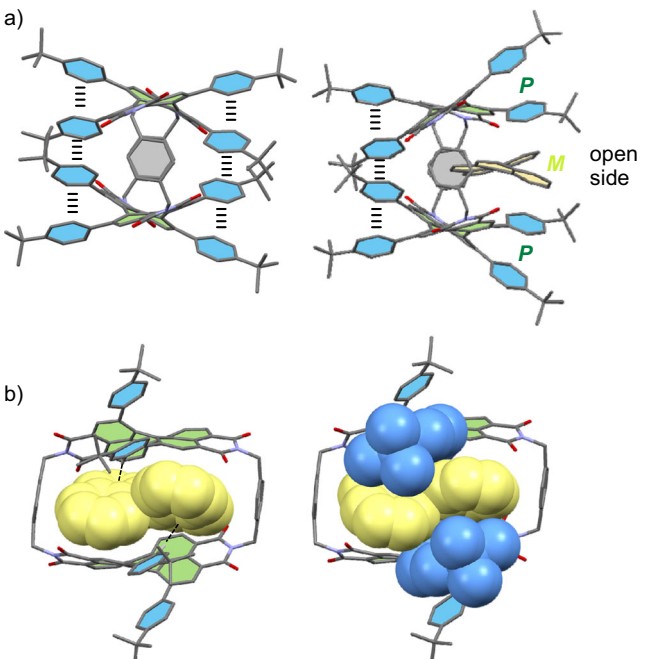

**Fig. 5 | DFT calculations.** DFT calculated structure of (**a**) **1-PP** and *M*-[5]helicene⊂ **1-PP** from the front view with indicated π−π stacking of phenyl substituents and (**b**) side view of *M*-[5]helicene⊂**1-PP** (bay substituents are partially omitted for clarity). The core of the PBI units and the bay substituents are highlighted in green and blue, respectively. The guest is highlighted in yellow. Hydrogen atoms are omitted for clarity.

## Structural insights from computational studies

In order to rationalize the observed binding affinities for the various guest molecules, we carried out DFT calculations for selected complexes under investigation (Supplementary Fig. 19). Pleasingly, whilst the guest-free structure of **1-PP** is of higher symmetry (point group: $C_2$ or $D_2$ if the *tert*-butly moieties are neglected) and characterized by the four *tert*-butylphenyl bay substituents being stacked at the periphery of both sides of the cyclophane, the calculated structures of all host-guest complexes reveal distorted geometries for the cyclophane host that are rather similar to the one observed in the solid state structure (with embedded solvent molecules) in our single crystal X-ray analysis (Fig. 2). Thus, the distorted nature of the cyclophane with an opening on one side as well as a rotational offset of the PBI units are apparent both in the crystal structure and in the computational models of the complexes (Supplementary Figs. 19, 20a). It turned out that the rigid planar guests, e.g. anthracene and pyrene, prevail in a flat geometry within the cyclophane cavity and abstain an adaption of their structure (Supplementary Fig. 19b, c) whilst the other guests are embedded in the host in non-planar geometries (Fig. 5a and Supplementary Fig. 19a, d, f). As the most important outcome of our calculations, we obtained computational support for our experimental data with regard to the complex with [5]helicene for which the opposite configuration compared to the host is indeed energetically preferred (Fig. 5 and Supplementary Fig. 19e).

Eventually, we see from the calculated structures that the distorted geometry of the PBI cyclophane host displaces especially the larger guests like [5]helicene and perylene out of the center of the cavity, which is opened on one side into a cleft-like receptor suitable for substrate recognition (Fig. 5a and Supplementary Figs. 19f, 20b). Notably, all calculated host-guest complex structures show embedded guest molecules of opposite chirality that are preferred within the cavity according to DFT calculations and stabilized to some degree by CH·π interactions between the *tert*-butylphenyl bay substituents of the host and the π-cloud of the guest. Thus, the calculated complex structures reveal that the substrates circumvent the shape mismatch between the sterically demanding bay substituents and the helical structure of the substrate by replacing π-π by CH·π interactions, thereby favoring heterochiral ensembles (Fig. 5b and Supplementary Fig. 19a, d, f). ALMO energy decomposition analyses show that the relevant bay-substituents are responsible for 34% of intermolecular interactions in the *M*-[5]helicene⊂**1-PP** complex and, remarkably, for 54% of the electrostatic interactions (Supplementary Fig. 20c, d) while in the *P*-[5]helicene⊂**1-PP** complex, the close bay-substituents account only for 27% of intermolecular interactions and for 42% of the electrostatic stabilization (Supplementary Fig. 20e, f). Accordingly, the unexpected configurational preference can be explained by the conformationally rigid *tert*-butylphenyl bay substituents that disturb the helical match of host and guest by a steric congestion with the embedded guest in the homochiral epimeric complex that reduces its thermodynamic stability and leads to the preferential formation of the heterochiral epimeric complex. In this cyclophane, the substitution pattern of the chiral PBI chromophore receptor site has therefore a strong impact on the enantioselectivity of guest binding. Thus, by means of such peripheral substituents, the preference of synthetic receptors for molecular recognition of chiral substrates can be switched from homo- to heterochiral epimeric complexes (Fig. 6).

## Further experimental support

Our conclusions drawn from the DFT structures are corroborated by the chemical shifts observed in the ¹H NMR titration study (*vide supra*) for the various protons as analysed in detail in Supplementary Fig. 41 (and text below). As an ultimate proof for the heterochiral molecular recognition, however, we can further provide the crystallographic analysis for a co-crystal grown from a mixture of *rac*-**1** and 1,1′-biphenyl (Fig. 7a, for details see the Supplementary Information).

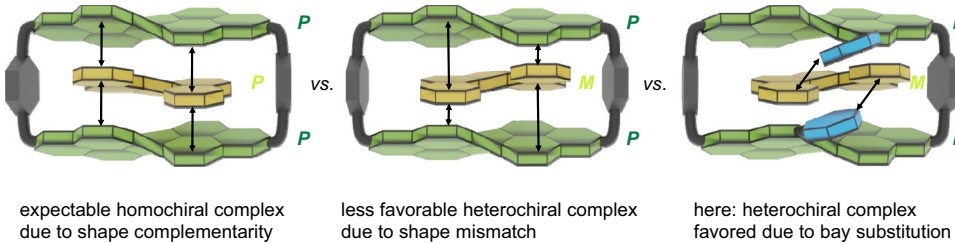

expectable homochiral complex
due to shape complementarity

*vs.*

less favorable heterochiral complex
due to shape mismatch

*vs.*

here: heterochiral complex
favored due to bay substitution

**Fig. 6 | Encapsulation of heterochiral guests by PBI cyclophanes.** Schematic representation of the expected energetically preferred homochiral complex and the less favourable heterochiral complex in the absence of bay substituents as well as the experimentally observed favored heterochiral complex due to the steric impact of the peripheral bay substituents.

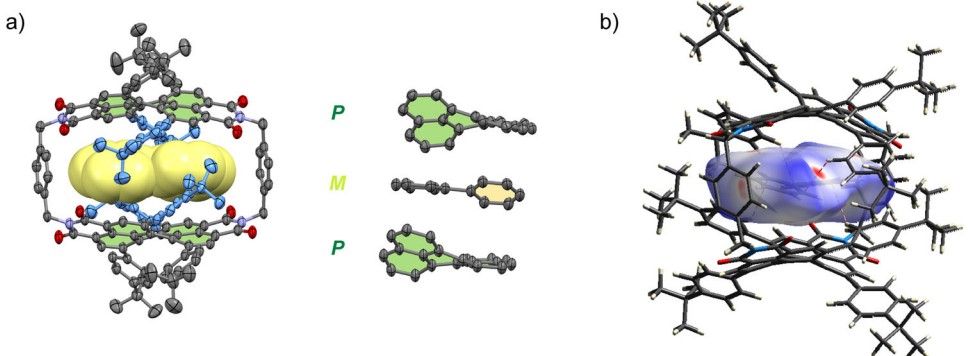

**Fig. 7 | Single crystal X-ray analysis of a heterochiral complex. a** Side view of molecular structure of *M*-1,1′-biphenyl⊂**1**-*PP*, obtained from single crystal X-ray analysis (surrounding guest molecules are omitted for clarity), corresponding perylene units with the encapsulated guest in order to visualize the heterochiral host-guest stacking. The perylene core of the PBI unit and the relevant bay substituents are highlighted in green and blue, respectively. The biphenyl guest is highlighted in yellow. Hydrogen atoms are omitted for clarity (the thermal ellipsoids are set to 50% probability). **b** Hirshfeld surface analysis to visualize CH-π interactions between the substituents and the guest (close C···H contacts shorter than 2.65 Å are highlighted by dashed lines).

The structural elucidation revealed not only the expected encapsulation of the 1,1′-biphenyl guest within the cyclophane cavity, but also this 1:1 complex being embedded in a 1,1′-biphenyl matrix and thus 4.5 equivalents of 1,1′-biphenyl per cyclophane molecule in the crystal (Supplementary Fig. 23a, b and text below). Accordingly, no solvent molecules are present in this co-crystal. The cyclophanes pack in a similar fashion as observed for the pure racemic host in chloroform (*vide supra*). More importantly, the complex structure clearly supports the DFT calculation for this complex (Supplementary Fig. 19a), as an opposite twist of the guest compared to the surrounding PBI units of the host is apparent. We would like to note that free 1,1′-biphenyl itself is configurationally unstable and therefore achiral at r.t. However, what we consider here as opposite configuration is the opposite twisting along the phenyl-phenyl single bond compared to the twist of the perylene core of the host as a result of optimized host-guest interactions. Hence, the solid-state structure proves the preference for the formation of heterochiral complexes for this chiral cyclophane. Notably, the structure found in the single crystal reveals, similar to our calculated models, that the sterical congestion imparted by the peripheral *tert*-butylphenyl substituents are responsible for this unusual stereochemical preference, which becomes especially apparent in the space filling model (Supplementary Fig. 24) and in our Hirshfeld surface analysis of the guest molecule depicted in Fig. 7b, revealing close CH···π contacts between biphenyl and the aromatic protons of the bay substituents.

## Discussion

Substrate-specific molecular recognition is a hallmark in supramolecular host-guest chemistry. Herein, we presented a chiral cyclophane composed of two core twisted PBI chromophores, which provided unprecedented insights into the molecular recognition of planar and in particular non-planar aromatic guest molecules. Titration studies revealed that the binding affinity of a series of polycyclic aromatic hydrocarbons towards the cyclophane is not only dependent on the number of π-electrons that the guest offers, but also on its geometrical adaptability. For chiral [5]helicene, unexpectedly, our studies revealed a preference for the formation of the heterochiral epimeric complex, where the helicity of the PBI units and the guest molecule have the opposite helical turn. This is a rare example of a preferential heterochiral guest recognition, which could be tracked by time-dependent NMR studies and rationalized by DFT calculations, indicating a general preference for the complexation of guests with an opposite configuration by the given host cavity. Our conclusions could be corroborated by a co-crystal for one of the complexes and the analysis of the chemical shifts in the NMR spectra of the complexes. From these data we are able to generalize our unprecedented results and to suggest a design principle for molecular receptors for heterochiral recognition based on the smart functionalization of the central recognition site with peripheral substituents that modulate the binding affinities by their sterical bulkiness and thus by repulsive forces and additional noncovalent interactions to the substrate.

## Methods

### General

Chemicals and solvents were purchased from commercial suppliers and used without further purification. Precursor **2**[46], [5]helicene[52] and *N-Boc*-4-(aminomethyl)benzylamine[53] were synthesized according to literature known procedures (see Supplementary Information). The separation of the enantiomers of [5]helicene was achieved by chiral HPLC (DCM/*n*-hexane 3:7, flow rate 6.5 mL/min) with enantiomeric excess of >95%. Analytical HPLC was perfomed on a JASCO device (PU 2080 PLUS) with a diode array detector (MD 2015), equipped with a

ternary gradient unit (DG-2080-533). Semipreparative HPLC was performed on a JAI LC-9105 using a Trentec Reprosil-100 Chiral-NR 8 μm-column for chiral resolution. For the Gel permeation chromatography (GPC) we used a Shimadzu Recycling GPC-System (LC-20AD Prominence Pump; SPDMA20A Prominence Diode Array Detector) with three or two preparative columns (Japan Analytical Industries Co., Ltd.; JAIGEL-1 H, JAIGEL-2H and JAIGEL-2.5 H) in chloroform (HPLC grade, stabilized with 0.1% EtOH) with a flow rate of 6.5 or 5.0 mL/min. NMR spectra were recorded on a Bruker Avance III HD 400 spectrometer at 295 K. Chemical shift data are reported in parts per million (ppm, $\delta$ scale) and are calibrated to the residual proton (for proton NMR) in the solvent (CDCl$_3$: $\delta = 7.26$; C$_2$D$_2$Cl$_4$: $\delta = 6.00$, CCl$_4$/C$_7$D$_{14}$ 3:1 (v:v): $\delta = 1.62$ (most downfield shifted signal)) or to the carbon resonance (CDCl$_3$: $\delta = 77.16$). High-resolution ESI TOF spectra of all literature unknown compounds were acquired on a Bruker Daltonics microTOF focus spectrometer.

### Optical spectroscopy

All spectroscopic measurements were carried out under ambient conditions. The UV/vis absorption spectra were recorded on a JASCO V-770 or V-670 spectrometer equipped with a PAC-743R Peltier for temperature control. CD spectroscopic measurements were performed with a JASCO J-810 spectropolarimeter equipped with a Jasco CDF-426S Peltier temperature controller or with a customised JASCO CPL-300/J-1500 hybrid spectrometer. CPL spectra were recorded with a customised JASCO CPL-300/J-1500 hybrid spectrometer. Fluorescence spectroscopic measurements were performed on an Edinburgh Instruments FLS981 fluorescence spectrometer. The quantum yields were determined under highly dilute conditions (A <0.05) relative to Oxazine 1 ($\phi_F = 11\%$ in ethanol)[54] as a reference compound.

### DFT calculations

Energy-minimized structures were obtained by DFT calculations (Gaussian 16)[55] on the B3LYP-D3/6-31 G(d) level of theory. Frequency calculations at the same level of theory were performed on all optimized structures to confirm them as equilibrium structures. Second-generation ALMO energy decomposition analysis[56] was applied to the optimized structures of M-[5]helicene⊂1-PP and P-[5]helicene⊂1-PP to decompose the non-covalent interaction energy ($E_{int}$) into electrostatics ($E_{elec}$), dispersion ($E_{disp}$), polarization ($E_{pol}$), charge-transfer ($E_{CT}$) and Pauli repulsion ($E_{Pauli-rep}$) contributions with the B3LYP-D3/6-311 G(d) level of theory. For the calculation of the CD spectra by TD-DFT we used the CAM-B3LYP functional and 6-31 G(d) as a basis set (scrf: chloroform). We note that our TD-DFT calculations do not consider vibronic coupling.

### Single crystal X-ray analysis

The diffraction images for X-ray crystallographic analysis of rac-1 were collected on a Bruker D8 Quest Kappa diffractometer with a Photon II CMOS detector and multi-layered mirror monochromated Cu Kα radiation. Single crystal X-ray diffraction data for 1,1′-biphenyl⊂rac-1 were collected at the P11 beamline at DESY. The diffraction data were collected by a single 360° ϕ scan at 100 K. The diffraction data were indexed, integrated, and scaled using the XDS program package[57]. In order to compensate low completeness due to single-axis measurement two data sets were merged using the XPREP program from Bruker[58]. The structure was solved using SHELXT[59], expanded with Fourier techniques and refined using the SHELX software package[60]. Hydrogen atoms were assigned at idealized positions and were included in the calculation of structure factors. All non-hydrogen atoms in the main residue were refined anisotropically. Disordered solvent molecules were modelled with restraints using standard SHELX commands DFIX, SAME, SADI, DELU, SIMU, CHIV, ISDOR, and RIGU. Because the refinement for 1,1′-biphenyl⊂rac-1 was not stable presumably due to pseudo-symmetry between two crystallographic

isomers, the DAMP command of SHELX was applied to converge refinement. Hirshfeld surface analysis[61] was done using Crystal Explorer 21.5 on the X-ray crystal structure of 1,1′-biphenyl⊂rac-1.

### Complexation studies

For the titration experiments, a mixture of PBI cyclophane rac-1 and the corresponding guest in excess was titrated to a solution of the pure cyclophane in the same solvent (or solvent mixture) of the same concentration to keep the host concentration constant during the experiment. The UV/vis and NMR titration data were fitted to a 1:1 binding model[62,63]. In addition we also carried out a global fit analysis with the program bindfit[64] in a suitable spectral range. For the titration studies with enantiopure [5]helicene, several host-guest solutions of different stoichiometric ratios were prepared and measured immediately after guest dissolution to avoid kinetic effects due to guest racemization.

## Data availability

Crystallographic data have been deposited with the Cambridge Crystallographic Data Centre as supplementary publication no. CCDC 2207897 and CCDC 2207898. Copies of the data can be obtained free of charge via https://www.ccdc.cam.ac.uk/structures/. The authors declare that all other data supporting the findings of this study are available within the article and its Supplementary Information file. Source data (UV/vis titration studies) are provided with this paper. Supplementary data 1 contains the cartesian coordinates of the calculated structures.

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

## Acknowledgements

We thank the Fonds der Chemischen Industrie for a Kekulé Fellowship for M.W. We acknowledge DESY (Hamburg, Germany), a member of the Helmholtz Association HGF, for providing experimental facilities at PETRA III (proposal No STP-20010387). We thank Dr. Helena Taberman for assistance in using P11. The CPL/CD hybrid spectrometer was funded by the Deutsche Forschungsgemeinschaft (DFG, German Research Foundation) - Projektnummer 444286426. This publication was supported by the Open Access Publication Fund of the University of Wuerzburg.

## Author contributions

F.W. conceived and supervised the project. M.W. carried out the synthesis, spectroscopic and binding studies and performed the DFT calculations. K.S. performed the X-ray analysis. M.W. and F.W. co-wrote the manuscript.

## Funding

## Competing interests

The authors declare no competing interests.
