## [Peer Review File · Nature Communications]

Preferential Molecular Recognition of Heterochiral Guests within a Cyclophane ReceptorREVIEWER COMMENTS

Reviewer #1 (Remarks to the Author):

The manuscript by Weh et al. provides an interesting example of a cyclophane receptor and its interactions with guests. The study has been carefully performed and contains some really nice results, notably the behavior toward [5]helicene, which in my view, merit publication in a high-profile journal. In the present form, however, the work contains some ambiguities and editorial issues (detailed below), which require attention before a final decision can be made by the editors.

General comments

The usage of terms “homo-/heterochiral,” “self-discrimination” and “self-recognition” should be reconsidered by the authors. In their primary sense, they refer to either identical molecules or true enantiomers. The present work uses these terms in a broader sense, by relying on structural analogies, e.g. the presence of similar stereogens. This causes two problems: first, homo-/heterochirality is claimed between structurally quite distinct molecules (e.g. 1-PP and biphenyl), second, self-discrimination/recognition is proposed to occur between chemically distinct molecules (“self” implies they should be identical or truly enantiomeric). I believe it should be possible to tell the same story using more rigorous terminology.

The selectivity of cyclization toward homochiral cyclophanes is of interest. One can imagine that cyclization of 3 into 1 is not feasible with heterochiral subunits, however, the first imidization should not be affected (and may be one of the factors lowering the yield of rac-1). Further oligomerization could in principle lead to larger cyclophanes; have they been observed?

Specific comments

Title: “Preferential Molecular Recognition of Heterochiral Guests”

Recognition is preferential by definition. I am not sure if the paper actually shows a case of true recognition: the binding of [5]helicene seems to be indiscriminate (if not, binding constants should be given for both enantiomers), and the differences observed in Fig. 4 appear to stem from the different initial composition of the mixture (it is difficult to be fully conclusive here, because the NMR is in the fast exchange limit). Over time, [5]helicene complexes converge toward a racemate-like spectrum, i.e. it is not clear if any preference is actually observed. Other guests are effectively achiral, and what is observed is a chiral induction event rather than recognition.

“Homochiral self-sorting, also called narcissistic self-sorting”

Narcissistic self-sorting is a more general term.

“assemble in domains with their mirror image”

“in domains” is unclear here: how large is the domain? Note that the racemate can be seen as an infinitely large “domain” of this kind, and is not considered a case of self-discrimination

“enforces our imagination”

style

“leading to a quadruple π -stack”

the figure apparently shows four (double) stacks, which is not the same.

“distorted symmetry-broken geometry”

This is not clear without specifying the reference symmetry that is being broken according to the authors.

“(700–400 nm).”

If not a typo, the range could be reversed.

Fig. 3a

The layout combining datapoints and structures is difficult to read. The structures might preferentially be numbered and the structures shown in a separate chart, or the structures could be isolated from the graph and linked with it using guidelines. In any case, the data markers are too big and there is a color mismatch for the blue series (rings are filled with cyan).

The classification appears to be rather arbitrary: the three series coincide on the DeltaG scale and the compounds are divided in a way that does not seem to show any actual pattern or structural relationship.

A more general issue with this analysis is that it assumes that the number of double bonds should give some sort of correlation. Is it known with certainty, that the largest guest can entirely fit into the cavity, so that the entire pi system is engaged into the interaction with the host? As an additional safety measure, the authors could check if the interaction with the 3a reference is negligible for the largest guests, to completely rule out external binding.

Errors determined for Ka should be converted into esds for DeltaG values, and should be indicated in the table and if they are significant – also in the figure (as error bars).

“However, it should be noted that the central ring of perylene is less aromatic with long carbon-carbon bonds and that already the hydrogen substituents in the bay area of perylene suffer from some crowding. Thus, little energy is needed for a rotational twist of the two naphthalene subunits in a propeller-like structure for the perylene scaffold.”

This is a rather detailed structural hypothesis, which is not supported by available data. Modeling the host–guest complexes (rather than isolated guests, as in Fig. 3b) could shed light on the origins of stabilization, especially if a distortion–interaction analysis is performed. Such data could provide support for some other claims related to binding specificity

“twisted and rigid topology”

these are not topological attributes (in the sense used by the authors)

“Only pyrene with its particularly incongruous stiff geometry of the central naphthalene subunit”

This is also conjectural and lacks precision. What is the reference for “incongruity”? Why is pyrene considered “particularly stiff”? It is also a bit unexpected (though formally correct) to refer to a naphthalene subunit, since pyrene, with two Clar sextets, is more of a biphenyl subunits

“Although [5]helicene has compared to [4]helicene a lower affinity for the complexation by the cyclophane”

syntax (or punctuation)

“1-PP is of higher symmetry”

All symmetries should be specified.

“four tert-butylphenyl bay substituents”

The system seems to possess 8 such groups. Please check

“highly strained geometry of the PBI cyclophane”

It is not clear what strain is meant here: the intrinsic strain of the host (is it known?) or the extra strain incurred on binding (again, a distortion–interaction analysis would be helpful)

Reviewer #2 (Remarks to the Author):

This is an innovative and exciting new piece of work by Würthner and co-workers that describes the encapsulation of dynamically chiral guests by a chirally locked host to form heterochiral supramolecular complexes, the opposite of the expected homochiral molecular recognition. Due to the norms of chiral complementarity, this is unusual and certainly a noteworthy result worthy of publication here. This has significance for the field of chiral molecular recognition and sensing, which as the authors state is an area of current interest in supramolecular chemistry, particularly for chiral pi-systems that are distinct from the classical three-point model that often works for hydrogen bonding systems. The results have wider importance to understanding chiral binding in biological systems. While previous work by the authors has seen them generate a configurationally locked chiral host by connecting two bay-strapped perylene diimides (ref. 22) this work is clearly differentiated by the use of four bulky bay substituents to lock the chirality of the host. These systems have been shown to be chirally locked, including by the authors (ref. 45), although Hofmann's earlier work on these systems could also be cited here. This new approach to chiral locking is still effective in creating a chiral cavity for testing guest binding.

This is a thorough and quantitative investigation into the structure, chiroptics and then binding properties of the novel macrocycle. The authors have done very well in using a variety of techniques effectively to deepen their understanding, particularly in the case of interpreting the more unusual heterochiral recognition. As such the experimental results presented here are in agreement with the author's conclusions.

This reviewer has a few minor points to address some queries or in some cases potentially improve clarity for the reader

1. From the XRD and DFT models of the macrocycle the authors state there is pi-stacking between the aromatic rings on the bay substituents (i.e., on Fig. 2c). Is there any evidence for this in solution, such as from UV-vis spectra of the macrocycle in comparison to the monomer?
2. In either the 'synthesis and characterisation' or 'structural' sections it would be important to state to readers that the PDI units are chirally locked, especially since the guests tested later can interconvert. Perhaps a simple variable temperature CD spectrum could be included to demonstrate this?
3. In Figure 3b, it would be helpful if the guests were labelled.
4. Please check Figures S36 and S37, some have missing parts (c, d etc) in the figure/caption. Also there is no red line needed for Fig. 36c due to the UV-vis not changing.
5. Do the authors have an explanation for the UV-vis titrations that do not generate an increase in red-shifted absorption? For example in Figs S25, S31 and S38 the only change is to S0-S1 and S0-S2 bands with no isosbestic points. While I do not believe this invalidates any conclusions here it would be interesting to include a statement as to why some guests do not produce the charge transfer absorption trait.
6. It might be valuable to add the K_a values for the chiral P and M [5]helicenes to Table 1, so that the reader can see the clear preference for M over P.
7. From the DFT/XRD structures for heterochiral complexes (helicene/biphenyl) is there any evidence for pi-stacking interactions between the phenyl substituents and the guest or are the interactions only CH-pi? It might also be helpful to label the CH-pi interactions on Fig. 5 as for Fig. 7b.

Once these points have been suitably addressed in revised versions of the manuscript and supporting information, this reviewer would highly recommend publication.

Reviewer #3 (Remarks to the Author):

The Authors presented an important manuscript on chiral host-guest systems showing the relevance of different "forces" on the final supramolecular adducts.

The work compare favourably with the state-of-art in this relevant field. Also, the experimental and modelling studies are appropriate.

I particularly appreciate the hoe tha authors highlighted the relevance of repulsive forces in the final assemblies. I suggest, as a minor revision, to emphasize this very relevant aspect of their findings.

Review Nature Communications manuscript NCOMMS-22-38141

We thank the reviewers for their constructive comments on our manuscript. In the following we address all suggested improvements by the reviewers. The corresponding changes are highlighted in the manuscript. The reviewer's queries are marked in blue and our response in black.

Point-by-point response to reviewer comments:

Reviewer 1

The manuscript by Weh et al. provides an interesting example of a cyclophane receptor and its interactions with guests. The study has been carefully performed and contains some really nice results, notably the behavior toward [5]helicene, which in my view, merit publication in a high-profile journal. In the present form, however, the work contains some ambiguities and editorial issues (detailed below), which require attention before a final decision can be made by the editors.

- 1. The usage of terms "homo-/heterochiral," "self-discrimination" and "self-recognition" should be reconsidered by the authors. In their primary sense, they refer to either identical molecules or true enantiomers. The present work uses these terms in a broader sense, by relying on structural analogies, e.g. the presence of similar stereocenters. This causes two problems: first, homo-/heterochirality is claimed between structurally quite distinct molecules (e.g. 1-PP and biphenyl), second, self-discrimination/recognition is proposed to occur between chemically distinct molecules ("self" implies they should be identical or truly enantiomeric). I believe it should be possible to tell the same story using more rigorous terminology.*

We thank the reviewer for the shared excitement on our recent findings and the careful proofread of the manuscript. In our opinion, the nature of these complexes is more similar to other supramolecular assemblies, which are stabilized by π - π stacking to different extents depending on the configurational match (Safont-Sempere et al., *J. Am. Chem. Soc.* **2011**, *133*, 9580: "self-recognition" of helically twisted PBIs), while in contrast most natural and many other artificial enantioselective host cavities often enable specific binding by more directional hydrogen bonding, for example (TPA model). However, in accordance with the suggestion of the reviewer, we have now dispensed with the term "self-discrimination" and "self-recognition" in the context of the formation of the host-guest complexes and merely referred to the analogy mentioned. The terms "homochiral" and "heterochiral" are appropriate in our view, since the host and the guests are structurally related as both of them exhibit axial chirality. In our recent work on the first inherently chiral PBI cyclophane (Weh et al., *Angew. Chem. Int. Ed.* **2021**, *60*, 15323), we observed the importance of homochirality (perfect shape complementarity – similar to the preferential homochiral dimerization observed by Safont-Sempere et al.) between distinct molecules, i.e. the host and a guest, if non-directional interactions dominate the binding. The preferential binding of guests with the opposite configuration should accordingly – to be consistent – be called heterochiral, which describes best our findings.

We also note that the term "homochirality" is not restricted to one and the same molecule, but is also used, for example, to describe the natural occurrence of exclusively L-amino acids (different molecules characterized by the same configuration of their central chirality).

- 2. The selectivity of cyclization toward homochiral cyclophanes is of interest. One can imagine that cyclization of 3 into 1 is not feasible with heterochiral subunits, however, the first imidization should not be affected (and may be one of the factors lowering the yield of rac-1). Further oligomerization could in principle lead to larger cyclophanes; have they been observed?*

Oligomerization is indeed always a side reaction for this type of cyclization (which we already observed for several other PBI cyclophane systems, see e. g. Spenst et al., *J. Am. Chem. Soc.* **2017**, *139*, 2014). When *rac-1* was purified by gel permeation chromatography (GPC), evidence for oligomers with significantly shorter retention times were also observed here.

- 3. Title: "Preferential Molecular Recognition of Heterochiral Guests"
Recognition is preferential by definition. I am not sure if the paper actually shows a case of true recognition: the binding of [5]helicene seems to be indiscriminate (if not, binding constants should be given for both enantiomers), and the differences observed in Fig. 4 appear to stem from the different initial composition of the mixture (it is difficult to be fully conclusive here, because the NMR is in the fast exchange limit). Over time, [5]helicene complexes converge toward a racemate-like spectrum, i.e. it is not clear if any preference is actually observed. Other guests are effectively achiral, and what is observed is a chiral induction event rather than recognition.*

The binding constants for the heterochiral and homochiral complex with [5]helicene as guest were determined and can be found in the Supplementary Information Fig. 42 in the binding studies section. The corresponding values of the Gibbs free energies are also given in the manuscript. The preference for one enantiomer is manifested in a very similar binding constant for the heterochiral host-guest mixture compared to the racemic mixture, while the homochiral complex formation is less favored (see also chemical shift Figure 4). Accordingly, in the racemic mixture, the host preferentially recognizes the guest of opposite configuration. The reviewer alludes here furthermore to the question of the mechanism, i.e. whether both enantiomers are bound and subsequently a conformational adjustment occurs ("induced fit") or whether the equilibrium is successively shifted by the binding of one enantiomer ("conformational selection"). This question cannot be answered unambiguously and generally for all guests (it could also be a mixture of both mechanisms). However, the preferred substrate recognition of the heterochiral guest, i.e. the formation of the heterochiral complex, is independent of these kinetic models.

- 4. "Homochiral self-sorting, also called narcissistic self-sorting"
Narcissistic self-sorting is a more general term.*

We agree with the reviewer, thank for his/her comment and changed the terms in the suggested fashion.

- 5. "assemble in domains with their mirror image"
"in domains" is unclear here: how large is the domain? Note that the racemate can be seen as an infinitely large "domain" of this kind, and is not considered a case of self-discrimination*

Introduction, including this sentence, has been revised.

6. *“enforces our imagination” style*

We changed this sentence.

7. *“leading to a quadruple π -stack”
the figure apparently shows four (double) stacks, which is not the same.*

The formation of such a quadruple π -stack (two stacks of four phenyl units per cyclophane) is actually better evident in Figure 5 (without guest/solvent molecules). In fact, however, a short π -distance between the substituents of the two chromophores (and not only between the substituents of one chromophore) can also be observed in the crystal structure. This can be seen particularly well in the side view of the cyclophane in Figure 2b.

8. *“distorted symmetry-broken geometry”
This is not clear without specifying the reference symmetry that is being broken according to the authors.*

The free host is according to proton NMR of D_2 symmetry, while the solid state structure reveals a C_2 symmetry due to solvent encapsulation. Symmetry was specified by giving the corresponding point group in the text.

9. *“(700–400 nm).”
If not a typo, the range could be reversed.*

The range is now reversed.

10. *Fig. 3a
The layout combining datapoints and structures is difficult to read. The structures might preferentially be numbered and the structures shown in a separate chart, or the structures could be isolated from the graph and linked with it using guidelines. In any case, the data markers are too big and there is a color mismatch for the blue series (rings are filled with cyan).
The classification appears to be rather arbitrary: the three series coincide on the ΔG scale and the compounds are divided in a way that does not seem to show any actual pattern or structural relationship.
A more general issue with this analysis is that it assumes that the number of double bonds should give some sort of correlation. Is it known with certainty, that the largest guest can entirely fit into the cavity, so that the entire π system is engaged into the interaction with the host? As an additional safety measure, the authors could check if the interaction with the 3a reference is negligible for the largest guests, to completely rule out external binding.*

We made a new suggestion for Figure 3 according to the criticism of the reviewer.

Fig. 3 Binding studies. a) Plot of Gibbs free complexation energies at 295 K versus the number of C=C double bonds of the guests' π -scaffold. The guests are classified in three groups: guests with a moderately bent or easily bendable structure that can adapt to the receptor (red), guests with a rigid planar structure (grey) and guests with a strongly twisted three-dimensional structure (blue). b) Energy minimized structures by DFT of selected guest molecules and c) chemical structure of the substrates which were employed as guests in the titration studies.

The classification into the three suggested groups is in our opinion reasonable. This becomes especially apparent for the guest series hydroxybiphenyl (strongly twisted as a result of the hydroxyl substituent), fluorene (rigid, planar) and 1,1'-biphenyl (flexible, moderately bent), where the Gibbs free energy continuously increases. The structural difference between 1,1'-binaphthyl and perylene lies essentially in the more limited ability of binaphthyl to adapt its shape. The same applies, for example, for benzoperylene and [5]helicene, while the guests marked in grey are not able to optimize their geometry upon binding not because of their three-dimensional structure but, on the contrary, because of their limited flexibility. Thus, while some degree of 3-dimensional cavity fit is necessary (impossible for rigid, planar PAHs), an overly bulky guest structure is a hindrance to efficient binding.

We added furthermore the suggested control measurement with reference **3a** and benzoperylene (see Supplementary Information Fig. 44). Upon the addition of approximately the same amount of equivalents of the largest guest to the reference compound as to the cyclophane host, we see from the UV/vis data that external binding can indeed be ruled out (or is negligible).

Supplementary Fig. 44 a) UV/vis spectra of *rac*-**3a** in CCl₄ at 22 °C ($c = 30 \times 10^{-6}$ M) upon the addition of benzo[ghi]perylene and b) the resulting plot of the absorption at $\lambda = 600$ nm.

11. Errors determined for K_a should be converted into esds for ΔG values, and should be indicated in the table and if they are significant – also in the figure (as error bars).

The errors given for K_a are the analytical errors from the local fitting with the 1:1 binding model. We added this information as a footnote in Table 1. We note that the resulting error for ΔG is very small (<1.5 kJ mol⁻¹; for most experiments <1.0 kJ mol⁻¹).

12. “However, it should be noted that the central ring of perylene is less aromatic with long carbon-carbon bonds and that already the hydrogen substituents in the bay area of perylene suffer from some crowding. Thus, little energy is needed for a rotational twist of the two naphthalene subunits in a propeller-like structure for the perylene scaffold.”

This is a rather detailed structural hypothesis, which is not supported by available data. Modeling the host–guest complexes (rather than isolated guests, as in Fig. 3b) could shed light on the origins

of stabilization, especially if a distortion–interaction analysis is performed. Such data could provide support for some other claims related to binding specificity

We refer here to one of our publications, which has investigated and clarified this issue with the help of calculations on a high theoretical level (Jiménez et al., *Chem. Sci.* **2014**, 5, 608). However, we thank the reviewer for his comment and provide accordingly a comparison of the distortion energy that is needed upon complexation for some of the guests (see below). For perylene (a), anthracene (b) and *M*-[5]helicene (c), we have compared the optimized free structure (green) and the structure within the cyclophane (grey). The difference in SCF energies (ΔE) shows that indeed only little energy is needed to distort perylene ($\Delta E = 3.4 \text{ kJ mol}^{-1}$) at the central benzene unit to significant dihedral angles of $\sim 14^\circ$ whilst for the other two examples similar or even higher energies are needed for more modest distortions.

Supplementary Fig. 21 Geometric and energetic differences (SCF energies) between the free substrate (green) and the complexed substrate (grey) by **1-PP**, given for a) perylene, b) anthracene and c) *M*-[5]helicene.

13. "twisted and rigid topology"

these are not topological attributes (in the sense used by the authors)

We thank the reviewer for this thoughtful comment. In fact, instead of "topology" the term "conformation" is probably more appropriate here. We have changed this accordingly.

14. *“Only pyrene with its particularly incongruous stiff geometry of the central naphthalene subunit”
This is also conjectural and lacks precision. What is the reference for “incongruity”? Why is pyrene considered “particularly stiff”? It is also a bit unexpected (though formally correct) to refer to a naphthalene subunit, since pyrene, with two Clar sextets, is more of a biphenyl subunits*

In contrast to biphenyl or perylene, the aromatic planes of anthracene or pyrene are rigid (biphenyl: twisting along the single bond; perylene: twisting along the "naphthalene-naphthalene" bonds possible). In the case of pyrene, however, there is the additional fact that no optimal π -stack between the perylene core of the chromophore and the substrate is possible ($\sim 1/4$ of the pyrene guest has no overlap with perylene even in the “best case”). Hence, the term “incongruous” refers to the fact that perylene and pyrene are not capable of optimal stacking (maximized π - π interaction) due to their different shape/geometry (see schematic depiction below). We have clarified this statement in the manuscript.

Pyrene is commonly described as “rigid” in literature (see exemplary citations below). However, we thank the reviewer for his/her thoughtful comment and refrain from using the term “particularly” to avoid confusion.

Chen, Y., Marszalek, T., Fritz, T., Baumgarten, M., Wagner, M., Pisula, W. & Müllen, K., *Chem. Comm.* **2017**, 53(60), 8474-8477:

“... since the rigid and bulky pyrene units...”

Chen, S. H., Chin, H. S., & Kung, Y. R., *Polymers* **2022**, 14(2), 261:

“...owing to the presence of rigid pyrene segments.”

15. *“Although [5]helicene has compared to [4]helicene a lower affinity for the complexation by the cyclophane”
syntax (or punctuation)*

Sentence was adjusted.

16. *“1-PP is of higher symmetry”
All symmetries should be specified.*

We thank the reviewer for this point of criticism. We specified now the symmetries by giving the corresponding point groups. The guest-free structure is of C_2 symmetry (D_2 without considering the *tert*-butyl moieties) while the complex structure is of C_1 symmetry (in any case).

17. *“four tert-butylphenyl bay substituents”*

The system seems to possess 8 such groups. Please check

In total, there are actually eight substituents. We have referred here to the stacking of four of these phenyl units and have now specified this information accordingly in the text. We thank the reviewer for his/her careful proofreading again.

18. *“highly strained geometry of the PBI cyclophane”*

It is not clear what strain is meant here: the intrinsic strain of the host (is it known?) or the extra strain incurred on binding (again, a distortion-interaction analysis would be helpful)

We thank the reviewer for pointing out this inaccuracy. The better term in this context is indeed “distorted”, which arises from the twisted nature of the chromophores in close proximity. We adjusted the terminology.

Reviewer 2

This is an innovative and exciting new piece of work by Würthner and co-workers that describes the encapsulation of dynamically chiral guests by a chirally locked host to form heterochiral supramolecular complexes, the opposite of the expected homochiral molecular recognition. Due to the norms of chiral complementarity, this is unusual and certainly a noteworthy result worthy of publication here. This has significance for the field of chiral molecular recognition and sensing, which as the authors state is an area of current interest in supramolecular chemistry, particularly for chiral pi-systems that are distinct from the classical three-point model that often works for hydrogen bonding systems. The results have wider importance to understanding chiral binding in biological systems.

While previous work by the authors has seen them generate a configurationally locked chiral host by connecting two bay-strapped perylene diimides (ref. 22) this work is clearly differentiated by the use four bulky bay substituents to lock the chirality of the host. These systems have been shown to be chirally locked, including by the authors (ref. 45), although Hoffmann’s earlier work on these systems could also be cited here. This new approach to chiral locking is still effective in creating a chiral cavity for testing guest binding.

We thank the reviewer for his/her kind and enthusiastic comments on our recent work. We added a citation of Hoffmann’s earlier work in our manuscript.

This is a thorough and quantitative investigation into the structure, chiroptics and then binding properties of the novel macrocycle. The authors have done very well in using a variety of techniques effectively to deepen their understanding, particularly in the case of interpreting the more unusual heterochiral recognition. As such the experimental results presented here are in agreement with the author’s conclusions.

This reviewer has a few minor points to address some queries or in some cases potentially improve clarity for the reader

1. *From the XRD and DFT models of the macrocycle the authors state there is pi-stacking between the aromatic rings on the bay substituents (i.e., on Fig. 2c). Is there any evidence for this in solution, such as from UV-vis spectra of the macrocycle in comparison to the monomer?*

The altered UV/vis spectrum of the macrocycle compared to the monomer is dominated by the coupling between the chromophores (that changes upon complexation of guest molecules, see discussion below). Since the phenyl substituents contribute little to the optical transitions, no direct conclusion on π - π interactions between them can be drawn from our data.

2. *In either the 'synthesis and characterisation' or 'structural' sections it would be important to state to readers that the PDI units are chirally locked, especially since the guests tested later can interconvert. Perhaps a simple variable temperature CD spectrum could be included to demonstrate this?*

We refer at this point to the recent work of our group on bay arylated PBIs (Renner et al., *Chem. Eur. J.* **2021**, 27, 11997 and Mahlmeister et al., *J. Am. Chem. Soc.* **2022**, 144, 10507), where we provide detailed studies on the racemization kinetics of PBIs bearing interlocked arene substituents in the bay positions (also phenyl and *tert*-butyl phenyl substitution).

3. *In Figure 3b, it would be helpful if the guests were labelled.*

We made a new suggestion for Figure 3 (see above).

4. *Please check Figures S36 and S37, some have missing parts (c, d etc) in the figure/caption. Also there is no red line needed for Fig. 36c due to the UV-vis not changing.*

We are grateful for this comment. We added the missing parts and the illustrations have been adjusted in accordance with this note.

5. *Do the authors have an explanation for the UV-vis titrations that do not generate an increase in red-shifted absorption? For example in Figs S25, S31 and S38 the only change is to S0-S1 and S0-S2 bands with no isosbestic points. While I do not believe this invalidates any conclusions here it would be interesting to include a statement as to why some guests do not produce the charge transfer absorption trait.*

Charge transfer bands are expected only if the HOMO is localized on one molecule and the LUMO on the other. However, most guests are "wide band gap" and accordingly both HOMO and LUMO are on the PBI units, see also Spent and Würthner, *Angew. Chem. Int. Ed.* **2015**, 54, 10165. Therefore, most changes are attributable to structural changes of the PBI units and a decrease of their coupling with each other due to an increase of distance upon guest complexation and not due to the host-guest interaction. Regarding the isosbestic points: Even though these are less pronounced in the experiments addressed, they do exist (especially in the region around 500 nm).

6. *It might be valuable to add the K_a values for the chiral P and M [5]helicenes to Table 1, so that that the reader can see the clear preference for M over P.*

We added now the corresponding binding constants for the heterochiral and homochiral complex with [5]helicene to Table 1 as a footnote. Moreover, we now also mention the binding constants explicitly in the text.

7. *From the DFT/XRD structures for heterochiral complexes (helicene/biphenyl) is there any evidence for π -stacking interactions between the phenyl substituents and the guest or are the interactions only CH- π ? It might also be helpful to label the CH- π interactions on Fig. 5 as for Fig. 7b.*

The relative orientation of the phenyl substituents and [5]helicene to each other suggests that there is no π - π interaction between these moieties. In order to specify the way in which the bay substituents contribute to substrate binding, ALMO calculations were performed (see Supplementary Information Fig. 20), which quantify more precisely the different contributions (dispersion, electrostatic etc.). The CH- π interactions were now labeled in Fig. 5 similar as for Fig. 7b.

Reviewer 3

The Authors presented an important manuscript on chiral host-guest systems showing the relevance of different "forces" on the final supramolecular adducts.

The work compare favourably with the state-of-art in this relevant field. Also, the experimental and modelling studies are appropriate.

I particularly appreciate the hoe tha authors highlighted the relevance of repulsive forces in the final assemblies. I suggest, as a minor revision, to emphasize this very relevant aspect of their findings.

We thank the reviewer for his/her kind evaluation of our work and thank him/her for proof reading the manuscript. We have added one sentence in the final discussion to emphasize the relevance of repulsive forces.

REVIEWER COMMENTS

Reviewer #1 (Remarks to the Author):

The authors provided detailed responses to my queries and revised the manuscript where necessary. The revised figure looks better, and the distortion-interaction analysis is appreciated. I can now recommend acceptance. Very nice work!

Reviewer #2 (Remarks to the Author):

The authors have answered all of my comments appropriately and the changes they have made in response to my points and those of the other referees has significantly improved the manuscript. It has my strong recommendation for publication.